# Effectiveness of Network Relations in Poland during the Economic Crisis Caused by COVID-19: Interorganizational Network Viewpoints

**DOI:** 10.3390/ijerph20021178

**Published:** 2023-01-09

**Authors:** Aleksandra Sus, Michał Organa, Joanna Hołub-Iwan

**Affiliations:** 1Faculty of Management, Department of Management, General Tadeusz Kościuszko Military University of Land Forces, Piotra Czajkowskiego St. 109, 51-147 Wroclaw, Poland; 2Faculty of Management, Department of Strategy and Management Methods, Wroclaw University of Economics and Business, Komandorska St. 118/120, 53-345 Wroclaw, Poland

**Keywords:** coronavirus, pandemic, strategic changes, oncological networks, economic virus

## Abstract

The global pandemic triggered by the SARS-CoV-2 virus has caused marked changes in the economic landscape in essentially every branch of the economy. The pandemic has disordered lives across all countries in the world and also affected the public sector in interesting ways. Our empirical research, of which selected elements are presented in this paper, was conducted under pandemic conditions. This paper aims to identify the relationships between selected distinguishing features of the oncological interorganizational network (exchange, engagement, reciprocity) and determine their effectiveness under the conditions of the economic crisis caused by the COVID-19 pandemic. A side thread, which concludes the article, is the introduction of the category “economic virus” into management terminology; i.e., a set of factors causing economic crises with a microbiological genesis. Of particular importance are considerations regarding the uncertainty of the length and depth of the health crisis-related economic effects in financial markets and corporate decision making.

## 1. Introduction

This paper’s theoretical and practical considerations represent an attempt to identify changes in the macro- and micro-landscapes of modern organizations in the context of a global crisis. The COVID-19 pandemic has clearly influenced the economy on a global scale, contributing to a reduction in global economic growth in 2020 at an annualized rate of −3.4% to −7.6% [1]. However, the public sector has also been severely affected. The observations of the authors of the article and the research work conducted by the interorganizational oncology network [2,3,4,5,6], the beginning of which dates back to 2019 (the pre-pandemic period), led to several conclusions. These conclusions formed the basis for the research assumptions adopted in this article:The COVID-19 pandemic caused changes in all elements of the macroeconomic environment in Poland;Changes in the macroeconomic environment have affected the functioning of Polish public entities, including hospitals [7];The research conducted by networks of oncology hospitals in Poland indicates a low level of effectiveness for network relations, which means that, as three fundamental elements creating the effectiveness of network relations, exchange, engagement, and reciprocity among these centers have been disturbed;A new factor has emerged in the ecosystem of contemporary organizations, which the authors call the “economic virus”; i.e., a set of factors causing economic crises with a microbiological genesis that are not strictly material but have both material and immaterial effects (the social consequences of the pandemic). This factor does not only concern Polish organizations but has a global character.

The authors investigated these assumptions using theoretical and empirical materials and studies (the authors’ own research carried out as part of the interorganizational oncology network). A point that deserves clear emphasis is that the empirical research carried out here is the only practical analysis in the world of the effectiveness of relations in an oncology network operating under the conditions of a pandemic. Thus, these relations do not yet have a benchmark in post-pandemic conditions, which is a limitation of the research conducted. Research on these elements has been planned in both post-pandemic and international contexts as part of a project intended to be financed with external funding sources. Moreover, the considerations presented in the paper have also been included in the conceptual foundations of an interdisciplinary research project entitled “The effectiveness of relations in oncology interorganizational network in Poland”, conducted as part of the InterEkon project financed by the Ministry of Science and Higher Education. The scientific objective of this research project was to develop a model of factors conditioning the effectiveness of relations in an interorganizational network created by specialized medical teams in the so-called oncological network. 

This article consists of six logically connected parts. The first part provides an introduction to the analyzed subject matter. The second part describes the changes caused by the pandemic in the Polish macroeconomic landscape. The third part defines the Polish interorganizational oncology network, its elements, and the formal conditions of its functioning. The fourth part characterizes the effectiveness of the relations in this network according to studies in this area from around the world. The fifth part of the study discusses selected elements of the empirical research conducted in the years April 2020–February 2021. The article ends with the conclusions drawn from the research conducted and, on their basis, indicates the directions of development and the potential uses of (a) the research results and (b) the conclusions drawn from them in terms of their utilitarian and practical value for other (public and business) sectors of the economy in Poland and worldwide. Furthermore, in the sixth part, the authors focus on an attempt to define and describe the economic virus as an element that, until the pandemic, was not taken into account as a factor causing crises, which is a novelty for the analyzed topic. The article’s structure corresponds to the deductive procedure of the research, and its substance focuses on explaining phenomena from the general to the specific.

The next section describes the economic environment, which has changed significantly as a result of the pandemic. Thus, it introduces and discusses the context in which the consideration of changes caused by a biological factor has been placed.

## 2. Changes in the Macroeconomic Environment Caused by the COVID-19 Pandemic

A classic macroeconomic environment analysis identifies a constant set of elements, including political, economic, sociocultural, technological, environmental, and legal factors. PESTEL analysis is a simple but widely used tool for generating development and identifying those places in an organization’s general environment that may affect its functioning. This tool is widely known and used in strategic analysis. It is also a good way to identify critical elements in which risk, uncertainty, and the first signals of the need to change an organization’s structure can be seen. These changes can take the form of both threats and opportunities and comprise the foundation of the organization’s future strategy. Taking advantage of changes in a company’s environment increases the potential chances of success as opposed to simply surviving or resisting changes [8].

The changes in economic factors caused by the coronavirus pandemic primarily concern the increase in the unemployment rate. The unemployment rate in Poland in September 2020 was 6.1%, an increase of one point (5.1% in September 2019) compared to the same time in 2019, according to the Central Statistical Office in Poland. This increase is not very high, especially compared to 2004 when the unemployment rate in Poland was 19.1%, but this is only the beginning of the economic changes caused by the coronavirus pandemic in this country, especially against the context of people returning from abroad, who could eventually number as many as 200,000 to 0.5 million [9]. Data on the level of unemployment in Poland is further undermined by the fact that, in accordance with EU coordination rules, it is not permitted to claim unemployment benefits if the last place of work was not Poland. An exception to this rule is when a person applying for unemployment benefits proves that their place of residence was in Poland while working abroad. This, in turn, is somewhat impossible to prove. In this context, the number of people who are registered as unemployed will be significantly lower, as it makes no sense to register with the relevant office if one cannot collect unemployment benefits [10]. Legal regulations result in the falsification of data and misleading assessment of the situation in Poland. In addition, the economic downturn—or, rather, its dramatic halting as a result of the government’s actions in the face of this biological threat—is also a factor of a political nature. The businesses that are closing down exponentially are mainly restaurants, gyms, cinemas, theatres, spa centers, physiotherapies, and beauty and hairdressing salons. In March 2020—i.e., at the beginning of the pandemic in Poland—42,000 companies closed their operations. Over the next three months, this trend began to be reversed due to the Polish government’s assistance in the form of the anti-crisis shield, reaching a level where another 16,000 companies gave up their business in September 2020. It is worth mentioning that Polish organizations are predominantly SMEs, which constitute 99.8% of the total number of enterprises, and micro-enterprises are the most numerous group among them (96.5%, 2.0 million). In addition, inflation is on the rise: prices in Poland have increased by 3.2%, including the prices of food and non-alcoholic beverages rising by 2.7% and the price of electricity by 4.6% (September 2020 versus the same month in 2019). According to forecasts, a similar rate of change in food prices and fuel prices should be expected [11].

In the area of sociocultural factors—i.e., the prevailing religion, traditions, and customs, including shopping habits, health awareness, the prevailing family structure, and the society’s attitudes toward foreign and domestic goods—it is worth noting the tragic changes in family relationships caused by the global pandemic: divorce rates are increasing year on year (in 2018, the number of divorces was, on average, 327 per 1000 marriages, and in 2019, it increased to 356; this ratio is increasing in cities in particular (from 393 to 423), while in rural areas, it has increased from 215 to 242), which—from a sociological point of view—has pejorative implications. There is growing aggression in society and conflicts between family members; consequently, children are being brought up in patchwork families without lasting and solid moral foundations. This is undoubtedly an opportunity for service organizations in the field of psychological and medical support (as well as pharmaceuticals, such as antidepressants) and legal aid, as well as dating sites, social media, and brothels, to grow spectacularly. The current global economic crisis is reshaping the IT industry, which is reflected in the technological environment.

At the same time, sales in stationary shops are falling due to the existing restrictions on the numbers of people in shops and the fear of leaving home. Consequently, this has necessitated changes in the employment structure and redundancies among employees hitherto working in shopping centers. A natural part of any crisis is an increase in unemployment, and in the current situation, its consequences will be changes in people’s shopping habits all over the world.

It is also interesting to note that coronavirus is particularly dangerous to people of what is known as old age; in other words, over 60 years of age. According to the Ageing Report: Economic and Budgetary Projections for the EU-27 Member States (2008–2060), the aging of society is a fact [12] that will have a direct financial impact on the public finance system by increasing the costs of health care and care for the elderly. According to the report, between 2007 and 2060, the cost of health care will rise from 4.0% to 5.4% of GDP, and the cost of care for the elderly will increase from 0.4% in 2007 to 1.1% in 2060. Total expenses in the area of health and social policy will increase by 2.1% of GDP by 2060, which will equate to around EUR 15 billion in additional spending from the state budget each year. These expenditures are a direct result of the spread of civilization diseases. They may decrease considerably due to the high mortality among the elderly owing to coronavirus infection. This should result in significant changes in the document we have quoted. With regard to socio-cultural factors, it is necessary to mention the increased interest in self-treatment, which has given rise to the possibility of making decisions about one’s health, not only in the context of self-medication but also in the sense of taking care of health according to the English notion of “self-care”, which refers to all the actions taken by a person to improve and restore health and to treat and prevent the diseases of civilization. Thus, the nutraceutical and dietary supplement industries are likely to increase their turnover in light of the economic changes caused by the pandemic. Factors that should also be taken into account include:-The increased interest in healthy living, eating, and sporting activities;-Prosumption; i.e., taking account of customers’ needs in the organoleptic, aesthetic, and content-related qualities of the services and products on offer;-The move away from the californication of needs; i.e., the situation in which trade seeks to establish a new world culture, imposing opinions and feelings on customers and aiming to unify the tastes and preferences of buyers on a global scale by reducing the influence of regional preferences on production. In this context, regional goods and the sale of native products are becoming increasingly important: cheese, milk, vegetables, fruit, etc.

The technological environment is created with the available methods and technologies, the application of which enables the transformation of resources into products and services. The total change in the structure of the Polish economy is a consequence of changes in the world. In addition, in this area, rapid, discontinuous changes take place, which represent the genesis of technological transformations and act as their carriers. Discontinuities are cases of intensive technological change in action [8]. Technological discontinuities can be described as innovations that dramatically affect the development of industries, the anticipation of which—in the case of the pandemic analyzed here—was not possible, even with the use of sophisticated forecasting and planning tools [13]. The technological revolution is taking place through the use of disruptive technologies, the intensity of which is currently growing rapidly due to the need to adapt to the changes caused by the pandemic. In addition, the factors that are further enhancing the development of technology are:-Mass implementation of the home-office model by all global companies with a service character, including schools and universities;-Changes in the ways start-ups are financed in the form of crowdfunding, as exemplified by the Kickstarter platform, which in recent months has been joined by Polish, Slovenian, and Greek companies;-Use of the wisdom of the crowd as a source of innovation—the concept of crowdsourcing is a cheap and fast solution in many different industries (e.g., Lego, Dell, Starbucks, P&G) and, in the conditions of a pandemic caused by a biological agent, also a reliable one;-The establishment of cooperative cooperation based on the sharing economy, which, on the one hand, diminishes the need to implement innovation in structures, thus reducing costs and resulting in the establishment of cooperation with competing companies, but, on the other hand, reduces the chances of companies that use this model of acquiring a spectacular competitive position. Nevertheless, in times of crisis, reducing expenses is necessary and not a choice.

Political factors reflect how the government influences the way the economy operates, including the extent to which it financially supports enterprises, the tax policy it pursues, the stability of the political environment, and the foreign trade and social welfare regulations it implements. In the case of Poland, the anti-crisis shield currently covers two stages: from March 2020 to June 2020 and from November 2020 onwards. The first stage included the possibility of taking advantage of an exemption from paying social security contributions for three months and offers of non-refundable loans to micro-enterprises of PLN 5000.00 (EUR 1200) and standstill benefits, the amount of which depended on the amount of the decrease in business turnover. In addition, companies could benefit from subsidizing part of the salaries of employees to keep their jobs, and the size of this aid was made dependent on the decrease in turnover. In stage one of the anti-crisis shield, the industries that suffered from the most significant falls in turnover began to emerge, and the tourism industry was among them. The solution proposed by the Polish government was the introduction of so-called tourist vouchers of PLN 500 (EUR 120) for Polish families, with which they could pay for a stay in a hotel, guesthouse, agritourism farm, etc., in Poland.

The public entities that were the subject of this research also operate in the context discussed above. In the next part of the paper, we discuss the research sample; i.e., the elements building the only oncology network in Poland.

## 3. Characteristics of National Oncology Network in Poland—Description of the Research Subject

The development of the National Oncological Network (NON) in Poland was formally initiated by the provisions of an ordinance (regulation) in December 2018—the Regulation of the Minister of Health of 13 December 2018 on the pilot program of care for recipients within the oncological network. The catalyst of—and, at the same time, the factor integrating—the activities around this initiative is the implementation of a new, more effective model of oncological care in Poland, focused primarily on comprehensive, transparent, and patient-focused therapeutic management. One of the most significant changes is the introduction of a treatment coordinator individually assigned to cancer patients. Testing of the model path of conduct was initially planned in a pilot program, which, from the beginning of 2019, covered two voivodeships: Dolnośląskie (Lower Silesia) and Świętokrzyskie (Holy Cross Province). The pilot covered patients whose medical procedures involved one of the five most common types of malignant neoplasms; that is, diagnosis of a malignant tumor (cancer) of the prostate, ovary, colon, breast, or lung (which are marked with specific entity codes according to the ICD-10 classification: C18-C20, C34, C50, C56, and C61). The main aim of the pilot procedure concerned the improvement in the results of oncological treatment as a result of applying an innovative approach to patients, who would be diagnosed and treated much faster than before. 

Two documents updating the functioning of the National Oncological Network have been formally introduced that amend the regulation regarding the pilot program of care in the framework of an oncology network: -Regulation of the Minister of Health of 2 October 2019, which primarily introduced an updated definition of the oncological network and added two more voivodeships to the pilot program: Podlaskie voivodeship and Pomeranian voivodeship;-Regulation of the Minister of Health of 18 August 2020, which set the end date of the pilot program of the analyzed initiative as 31 December 2021.

The definition of the considered network was finally stated as [14]: “a structure operating in a given voivodeship, which consists of a voivodeship coordinating center and a cooperating university clinical center with cooperation centers of the 1st and 2nd level, which cooperate in the field of oncological care for the beneficiary covered by the pilot program”. In general, the National Oncology Network will, therefore, constitute a complex structure embracing all individual voivodeship sub-networks. According to the Regulation of the Council of Ministers of 17 October 2019 on the reorganization of the Oncology Center of the Institute of Maria Skłodowska-Curie in Warsaw and the granting of the status of a state research institute to the Institute [15], the coordinator of the network is presently the entity acting under the name “National Oncology Institute Maria Skłodowska-Curie–National Research Institute”. The documents mentioned above are consistent with the concept of the National Oncological Strategy established formally by the provisions of Resolution No. 10 of the Council of Ministers of 4 February 2020 on adopting a multi-annual program under the name of the National Oncology Strategy for 2020–2030 [16]. 

Determining the type of the oncological network under consideration boils down to the problem of differences between centralized and decentralized interorganizational networks [17,18,19,20,21,22,23,24,25]. For each voivodeship coordinator, there is a central unit coordinating the activities of a centralized subnet of entities. From the national perspective, the entire network is formally coordinated by the National Research Institute mentioned previously. The Ministry of Health additionally supervises the entirety of the network. However, at the level of individual voivodeships (individual voivodeship coordinators), there are often dependencies that correspond more to the specificities of decentralized networks; i.e., intensive exchange of resources and information both through formal and informal routes. As a result, the structure of the National Oncological Network in Poland resembles pyramids connected with cooperating vertices.

The team of authors from the Wroclaw University of Economics and Business, using an original tool for researching the effectiveness of network relations, conducted (in cooperation with a professional research agency—IMAS International) extensive research covering the nodes of the described network. The aim was to define and characterize the types of relationship ties linking individual units and to characterize the mechanisms ensuring the effectiveness of the entire system based on three main criteria: exchange, engagement, and reciprocity.

Finally, the research sample included 19 hospitals: specialist oncology treatment units located in all provinces and forming the National Oncology Network (NON) in Poland–presented in Table 1. The authors obtained 34 sets of complete responses from respondents, with individual nodes of the examined interorganizational network represented by one to three respondents.

The following sections of this article present the concepts used to investigate the effectiveness of relationships in an interorganizational network, which represented the central part of the theoretical research model. 

## 4. Research Assumptions about Relationship Effectiveness in an Interorganizational Oncology Network—The Essence of the Effectiveness of Network Relation 

Research on interorganizational relations has occupied both theoreticians and practitioners for over 25 years. The main focuses are the determinants and characteristics of relationships in networks and analyses of the phenomena of growth and development in systems that have evolved from diads to networks and are composed of more elements, creating different types of interorganizational networks [26,27,28,29]. In addition, two main strands of theoretical considerations are leading the way: (a) resource dependence and the interorganizational exchange perspective linked to it and (b) transaction cost theory, the basis of which is to be sought in economic issues. In both, one can find studies that focus on identifying the antecedents of interorganizational networks [30] and the results of networking [31], as well as the structures of networks and ways of managing network systems. In addition, the literature on interorganizational networks focuses on issues such as the advantages of networks over individual systems and, more recently, the ambidexterity and dynamics of network systems [32]. This section of the paper explains the essence of the effectiveness of the relationships studied in the oncology interorganizational network. 

Functioning in a medical network system generates a synergy effect, which is mainly associated with the elimination of structural, temporal, and geographical barriers to achieve a common goal. In the analyzed oncological interorganizational network in Poland, this goal was to develop a common model of treatment of cancer patients in the shortest possible time. This is the so-called rapid oncology pathway. This goal was to be achieved by transferring resources and through cooperation in medical functions. The interorganizational oncology network in Poland can, therefore, be defined as an open set of medical (oncology) facilities constituting the nodes of the network, the aim of which, working in partnership in a specific ecosystem [33] that is highly probable to be identical for each node, is to care for oncology patients [2].

In the public and non-profit sectors, there is a public interest motive that alters the goals and outcomes pursued by the interorganizational network in quite a significant way. The effectiveness of such a network focuses on achieving systemic goals instead of organizational goals, as is typical in the private sector. This happens even if the motivators for integration and cooperation are weak. When the situational context is analyzed—i.e., critical external groups, such as decision makers (e.g., the Ministry of Health, the National Health Fund, research units) and funding bodies (e.g., the National Health Fund, the Marshal’s Offices)—effectiveness is measured by the degree to which the assumed organizational and business goals are realized. Moreover, the effectiveness of an interorganizational network is connected to the integration of the organizations entangled in a given network. The argument justifying the higher effectiveness of integrated networks in the health-care sector is the increased availability of specialized healthcare services, ensuring continuity of care despite geographical dispersion. The coordination of an integrated system increases the likelihood that all necessary services are provided in the network and that clients have access to the specialists and the competencies they need [31].

Network effectiveness [34,35,36,37,38,39,40,41] in the public sector [42,43,44,45,46,47,48] is an entirely different category than in business arrangements due to resource shortages, high diversity, and the number of problems with network recipients (the category “customers” sounds inappropriate with regard to oncology network patients), as well as the limited functionality of specialist training and the fixed, low salaries of network staff [49]. Therefore, the financial efficiency criterion was eliminated from the research procedure. Attention was focused on non-material objectives; i.e., identifying those factors that influence a particular element of interorganizational network efficiency—namely, relationships. 

The effectiveness of the relationships in the interorganizational oncology network can be understood as the result of a combination of process and structural network characteristics that produces the optimal exchange [50], engagement [51], and reciprocity [52,53] between network nodes. The objects of interorganizational exchange, determining the effectiveness of the relationships, are resources, both in material and non-material terms, and the typology of exchange focuses on material and energy flows [54]. The effect of exchange is the ability to achieve higher efficiency than the efficiency of the individual—this is one of the motives for building interorganizational networks [55]. Engagement in a network involves a certain degree of interdependence between partners, which results from entanglement in the negative (risks) and positive consequences (benefits) of operating in network structures. Interdependence influences the coherence of the network and the cooperation between its elements, which is based on trust; i.e., the belief that the counterparty will fulfill its obligations [56]. The attribute of reciprocity, by contrast, is a critical element in the formal assessment of ties by the parties to a relationship, leading to its construction, modification, and also its disintegration. The components of reciprocity are the parties’ commitment, the negotiation processes in establishing the contract, and its execution by the parties [57].

The effectiveness of the relationships in this network was examined from structural and process perspectives. The first approach takes into account the importance and frequency of sharing and transferring resources: -information (data on patients; direct and indirect contacts, both medical and non-medical: pharmaceutical companies, medical equipment suppliers, and subcontractors; information about medical research results, medical records, formal and legal regulations, procedural solutions, contracts, and diagnostic-preventive solutions; information about applied diagnostic tests, targeted therapies, meta-analyses and other analyses, such as Big Data and statistics; VMAT irradiation solutions; knowledge of cancer immunotherapy; applied personalized vaccines);-human resources (exchange of medical staff; sharing of experts, surgeons, and doctors of different specializations: oncologists, pathologists, molecular biologists, laboratory diagnosticians, bioinformaticians, physicists, radiotherapists);-equipment (specialized medical equipment robots, radiology equipment, standards for histopathological diagnosis, tissue biomarker analyses);-drugs (information on opiates used and on cytostatic drugs and valproic acid-based drugs, preparation of cytostatic drugs);-and patients.

The second approach—the process approach—examines the implementation of primary and secondary activities in line with M.E. Porter’s value chain. Again, benefits are analyzed in terms of importance and frequency. In the area of auxiliary functions, the following potential benefits were identified:-Cooperation between project teams;-Exchange of good organizational practices, including management systems; for example, quality and management knowledge and administration;-Transfer of knowledge concerning professional development processes, motivation systems, evaluation, control and recruitment of staff;-Development of procedures for improving patient services;-Shared access to modern technologies and related procedures;-Transfer of knowledge on cancer immunotherapy vaccines;-Shared access to medical equipment suppliers, pharmaceutical companies, and administrative and other equipment;-Transfer of knowledge on how to create resource reserves and store medical supplies, on pre-operative procedures, and on monitoring of drug interactions;-Improvement of treatment methods and processes and prescription of the drugs used;-Exchange of information on post-operative care;-Sharing of ways of building public knowledge about cancer prevention [58,59,60,61], which—according to the opinions of this paper’s authors—is an especially important issue;-Identification of early warning systems for complications after medical procedures and treatment tolerance solutions.

This set of the beneficial effects of cooperation between organizations entangled in network arrangements in the oncology care sector is not closed. Rather, it constitutes an introduction to the extensive empirical research currently being conducted in Poland in order to extend research activities to Europe. 

The research subjects were organizations establishing an oncology network in Poland that—similarly to business organizations—pursue their strategic objectives in the macroeconomic environment. In this context, the distinguishing features of oncology network effectiveness were examined. The results of the research are presented in the following section.

## 5. Empirical Research on National Oncology Network in Poland

The empirical research on the National Oncology Network (NON) in Poland was conducted using an original survey questionnaire, which was sent to representatives of 19 specialist medical entities. The total number of respondents who participated in the study was 34. It is worth noting that, as mentioned earlier, the study was conducted under pandemic conditions from April 2020 to Feburary 2021, with research materials developed in the pre-pandemic period. Thus, unintentionally, a picture of the effectiveness of the relationships in the NON in Poland during the COVID-19 pandemic was obtained, which made it possible to observe the specificity of this system’s functioning under crisis conditions. 

The sampling method precluded the implementation of random sampling techniques due to the lack of a sampling frame. The research sample selection consisted of dividing respondents into layers in accordance with their co-participation in the oncology network unit, followed by random selection from within each stratum of the research units—people with decision-making capacities regarding exchange, engagement, and reciprocity in the oncology network. The results for the analyzed case cannot be generalized to the entire population due to the lack of representativeness of the research sample. The research was based on a purposive research sample. The principle of purposive in this research procedure concerned formal participation (employment) in a hospital unit operating in the oncology network and decision-making influence on its functioning (decision-making capacity).

The answers to individual questions of the survey questionnaire provided data on the cooperation between the entities in question in terms of exchange, engagement, and reciprocity. Considering the survey’s form, these were essentially the subjective assessments of individual respondents concerning certain aspects of cooperation, which were deemed to be reliable and credible. While describing selected elements of the research, the original numbering of the questions was used for the sake of maintaining consistency between the designations and the conducted analytical process, providing their full content each time. It is worth emphasizing that, when answering question no. 2 of the survey (Q2), respondents chose the units in the National Oncology Network (NON) with which relations were most important (question no. 2a), second most important (question no. 2b), and third most important (question no. 2c). In the subsequent part, only the relationships indicated as the most important (marked in question no. 2a) were considered. The adopted research problem was formulated “regressively”—the response variable was relationship effectiveness, while the explanatory variables were exchange, commitment, and reciprocity. For the purposes of this study, relationship effectiveness (the response variable) was considered in the context of the answers to one of the questions in the survey questionnaire; namely, question no. 28 (Q28): Do you believe that the amount of resources exchanged with a particular entity increases the effectiveness of your organization? Question no. 28 assessed the effectiveness of a particular relationship and related directly to the amount of resources exchanged between the NON entities. The authors focused on this question because the global crisis caused by the COVID-19 pandemic highlighted how vital resource sharing and proper resource allocation become in terms of ensuring the proper functioning of individual nodes in a network system, as well as the entire interorganizational network. Resource exchange in medical networks, including the oncology network under consideration, seems to play a crucial role in maintaining operational efficiency, especially in the context of a pandemic. This exchange concerns information, medical personnel, medicines, and necessary technical resources; e.g., various types of medical equipment, especially ventilators, oxygen resources, etc. Therefore, the authors of this article assumed that the key aspect of the NON’s functioning in the context of shaping the effectiveness of relations was, in crisis conditions, the widely understood resource exchange.

Subsequent survey questions were then assigned to the different domains of the relationship between NON entities; in this case, the explanatory variables (exchange, commitment, and reciprocity). Most of the questions listed in Table 2 contained multiple specific questions within them. Since the answers to the specific questions related to a given question were expressed on the same scale, the answers to the particular questions were summed when there was more than one question, thus obtaining an aggregate response to the question.

Ultimately, this article’s authors formulated the following question: is it possible to precisely calculate exchange, engagement, and reciprocity, and then, on this basis, determine whether they influence effectiveness? The answer to this question was sought through regression analysis; i.e., by trying to find the linear combination of the domain characteristics that correlated most strongly with effectiveness; that is, the answers to question Q28—or, more precisely, Q28a, since we are limiting ourselves to the relationship rated as the most important.

The exchange domain (EXCH) was analyzed first and defined by Q6_2a, Q7_2a, and Q8_2a (indicated in Table 3). These were correlated with each other, with Q8_2a correlating most strongly with the others. The answers to the remaining questions correlated weakly; i.e., the information carried by these questions was largely independent. Bold indicates significant correlations; i.e., those with an absolute value exceeding the critical value at *p* = 0.05.

Multiple regression analysis (Q28a—dependent variable; QP6_2a, Q7_2a, and Q8_2a—independent variables) indicated, in turn, that Q8_2a was marginally related to relationship effectiveness. However, an association with relationship effectiveness was shown by Q6_2a and Q7_2a, as indicated in Table 4.

However, it should be emphasized that, from a formal point of view, the regression model was not significant in this case. The “*p*” values for all variables were clearly above 0.05, and the calculated multiple correlation coefficient was R = 0.42. Therefore, we could only conclude that the regression function of the form EXCH = 0.88 × Q6_2a + 1.39 × Q7_2a + 0.32 × Q8_2a, which numerically determined that the level of exchange correlated with effectiveness (Q28a) at a level of 0.42, as well as indicating that Q8_2a made the least significant contribution to the correlation with effectiveness—it was, therefore, a relatively weak correlation.

Next, the focus was on the domain of engagement (ENG), defined here through Q6_1a, Q7_1a, Q8_1a, and Q11_1, in particular (described precisely in Table 2). In accordance with the previously adopted assumption, in the cases of Q6_1a, Q7_1a, and Q8_1a, we limited ourselves to analyzing the most important relations: the so-called first-order relations (hence the “a”). On the other hand, Q11_1—especially important for the assessment of engagement between the NON entities—was constructed in the survey in a way that required the inclusion of all relationships; that is, with the subjects of the first- (a), second- (b), and third-order (c) relations. This fact was taken into account in the interpretation of the results. Table 5 shows the correlations between the questions considered, with significant correlations (with an absolute value exceeding the critical value at *p* = 0.05) in bold. 

The analysis shown in Table 5 suggests that the relationship information carried by the answers to Q8_1a largely duplicated the information flowing from Q6_1a and Q7_1a. In contrast, the answers to Q11_1 brought additional information due to the lack of correlation with the answers to the other indicated questions.

On the other hand, multiple regression analysis, shown in Table 6 (Q28a—dependent variable; Q6_1a, Q7_1a, Q8_1a, and Q11_1—independent variables), indicated that the answers to Q6_1a had the strongest correlation with the effectiveness of the engagement traits (the coefficient in bold was significant at the level of 0.05). By far, the question least significantly associated with effectiveness was Q8_1a.

Concluding the analysis on the domain of engagement, it should be indicated that, in this case, the regression model was significant. The multiple correlation coefficient R = 0.63. At this level, the regression function (representing a numerical assessment of the degree of involvement) ENG = 0.47 × Q6_1a + 0.19 × Q7_1a − 0.04 × Q8_1a + 0.25 × Q11_1, indicating a correlation with effectiveness, and this correlation can be perceived as significant.

In the last step of this analysis, the reciprocity domain (REC) was analyzed, which, indicated in Table 2, was addressed by the following questions: Q11_2, Q21_1a, Q21_2a, Q22a, and Q24a. It is worth mentioning that, in the case of Q11_2 (similarly to Q11_1 above), due to the construction of the questionnaire, relations with the three most essential entities from the point of view of the entity represented by a particular respondent were taken into account. In the case of the remaining questions, only the relations with the most important entity indicated, the so-called first-order entity (marked “a”), were analyzed.

In the case of the reciprocity domain, the focus was on calculating the coefficients of the multiple linear regression, as shown in Table 7. An additional column with normalized coefficient values was introduced to facilitate interpretation, as the explanatory characteristics were expressed on different scales (ranges of values). The multiple correlation coefficient R = 0.59 (*p* = 0.028 < 0.05), which could mean a significant correlation between efficiency (Q28a) and the regression function REC = 0.47 × Q11_2 + 0.28 × Q21_1a + 0.21 × Q21_2a + 0.05 × Q22a − 0.06 × Q24a, numerically indicating a degree of reciprocity in the relationship.

In the analyzed case, the highest coefficients (normalized) indicated the features of the reciprocity of relations with the strongest relationships with effectiveness (expressed by the answers to Q28a). These features were the level of reciprocity (Q11_2a—the higher the reciprocity, the more it increased effectiveness) and the number of specialized units cooperating with each other (Q21_1a—the greater the number of units, the better the effectiveness). The number of administrative units cooperating with each other (expressed by the answers to Q22a and Q24a) slightly influenced the effectiveness of the relationship. 

The implementation of the presented research procedure made it possible to determine three numerical measures of relationship domains in the NON: exchange, engagement, and reciprocity. On the other hand, the level of involvement (assessed using the ENG function) was most strongly related to the level of effectiveness. It can be indicated that, in the case of the conducted research, engagement turned out to be crucial for the effectiveness of the relationships.

This means that the definition of relationship effectiveness adopted in the interorganizational oncology network became inaccurate under the conditions of panic and drama in the medical units related to the events caused by the COVID-19 pandemic. In such a situation, the effectiveness of the relationships in the network was focused on engagement, which, in turn, indicated the maximization of cooperation in the context of the organizational and management functions performed in organizations, as well as the validity and frequency of the relationships in the given categories. In other words, under the conditions of the crisis and related threats, the hospitals building the oncology network in Poland did not focus on mutual assistance and exchange in terms of medical equipment, drugs, etc., but on supporting each other in the area of procedural solutions that did not involve the need for direct contact. The interorganizational exchange and the directly related reciprocity were significantly reduced in the crisis conditions, which was perfectly natural, as each entity had to fight for its survival and that of its patients. In such a situation, the study only allowed for the identification of the critical feature of relationship effectiveness in the interorganizational oncology network in Poland under the crisis conditions caused by the COVID-19 pandemic, blurring the image of this effectiveness in quasi-stable conditions. This confirmed that the change in the macroeconomic landscape, discussed in the first part of the paper, was much broader in scope than has been presented. Indeed, changes have been recorded in every branch of the economy, whether the business or public sector, manufacturing or services. In the study described here, this finding was entirely coincidental. 

The investigations carried out by the authors aimed to introduce a new category into the terminology of the management sciences. A new factor has appeared, which has reorganized the functioning of all individual organizational units (service and production businesses), the public sector (hospitals, army, offices), non-profit organizations (foundations, associations), and interorganizational networks (the studied oncology network but also others: technological, IT, fuel clusters, etc.). This factor is an economic virus, which the last part of the article attempts to define. It also presents conclusions.

## 6. Conclusions

The study of the source materials showed that the COVID-19 pandemic caused changes in all elements of the macroeconomic environment in Poland. These transformations affected not only business organizations but also those of a public nature. Our research on the effectiveness of relations in an interorganizational oncology network (the National Oncology Network in Poland), unintentionally started and completed during the pandemic, showed that entities entangled in the network—the nodes of the network—did not focus on strengthening the effectiveness of relations in this network during the crisis by initiating exchange and related reciprocity. According to the results of the research, the effectiveness of network relations was residually maintained, and its only distinguishing feature was involvement, which should be understood as cooperation between nodes in the network in the context of the organizational and managerial functions performed in organizations, as well as the importance and frequency of relations in the given categories.

As a result of the analyses conducted, a new factor in the macroenvironment of organizations was diagnosed, which the authors call an “economic virus”. This is a factor of microbiological origin causing economic crises that does not have a strictly material character but causes effects of both a material and immaterial nature (the social consequences of the pandemic).

In summary, the empirical research presented here concerned the effectiveness of relationships in an interorganizational oncology network. The research was conducted under the conditions of the economic crisis caused by the microbial threat and the related COVID-19 pandemic. The article does not present research results on the same issue under quasi-stable conditions, as the economic situation has not changed since April 2020. Therefore, it was impossible to empirically benchmark the effectiveness of the relationships in post-pandemic conditions. The points that conclude our discussion are the two main observations:The effectiveness of the relationships in an interorganizational oncology network—i.e., in a strictly public entity—lies primarily in involvement in the area of organizational and management functions;There is a new element to be considered in the description of macroeconomic environments (in addition to the political–legal and social aspects of macroeconomic environments), which is the microbiological environment and the economic virus associated with it.

In light of work in the biological sciences, an economic virus can be defined as an agent that does not have a cellular structure. Thus, it is not a living organism but one whose emergence triggers significant changes in the structure of existing economic systems. The metaphor of the black swan—representing sudden, unexpected, and unfavorable economic phenomena—has become well-established in the global literature. However, just as black swans occur with a certain frequency, economic crises arise with a certain probability. In the coronavirus case, the black swan [62] was no longer a metaphor but a perfect example of a situation that disorganized the functioning of the business and public sectors in each of their dimensions, whether economic, socio-cultural, political–legal, or ecological.

We understand the economic virus to be an unprecedented factor, unlike any other phenomenon so far studied in economics. However, we remain in the circle of metaphors taken from the natural sciences because, in nature, a virus is an inanimate creature capable of replicating only in an infected organism. Each virus is actually a new phenomenon with its own variability (it mutates). Therefore, we also propose a new science, the aim of which would be to systematize and develop knowledge on the impact of these new, unique, and unpredictable microstructures on the global economy, under the name of “virus economics”, a combination of the words “virus” (Latin: *virus*—“poison” or “venom”) and “economics” (Greek: *oikos* meaning “house” and *nomos* meaning “rules”). It would, therefore, be a science of the rules governing microbial crises. Our proposal is based on the idea of microstructures interacting with macroenvironments and macrosystems. We do not limit our considerations only to the COVID-19 virus and its impact on the economy because, as an analysis of the topic shows, the range of zoonotic viruses is expanding significantly, and they are crossing species barriers faster and more efficiently. However, we signal the readiness of the management and quality sciences to reflect on contemporary economics, which, being so strong and developed, has proven to be fragile and susceptible to this phenomenon. Similarly to the pathophysiology related to the high pathogenicity of coronaviruses [63], which is not fully understood, the impact of economic viruses on the various branches of the economy and related sciences is only just being recognized.

The economic virus that we have thus defined is the factor that caused the total collapse of the world economy, the boom in which lasted uninterrupted until February 2020 following the 2009 crisis. Neither armed conflicts nor acts of terrorism have slowed down economic growth. From observation and analysis of existing factors, it appears that the panic around the world caused by the coronavirus (the actual threat of the disease caused by the virus) brought about changes with effects that will have long-term significance for the Polish and global economies. The microbiological threat, which serves us as a metaphor for a new, hitherto non-existent primary (base) factor of an economic nature, today already has much more significant consequences for each branch of the global economy than those described in the study.

Research on the concept of the economic virus in the longer term can be carried out on an international scale, primarily for comparative purposes regarding the impacts of this identified phenomenon on the economies of selected countries (e.g., Germany, the Czech Republic, and France due to their relative proximity to Poland). Each of these countries developed a country-specific basis for its responses and was “marked” by the pandemic experience (as a biological phenomenon), generating specific conditions for conducting business and non-business activities. The authors plan to develop a universal comparative matrix taking into account selected macroeconomic indicators (primarily changes in GDP, unemployment rates, and inflation rates, utilizing several stages; e.g., “before the pandemic”, “during the pandemic”, “post-pandemic period”), the long-term monitoring of which would be intended to facilitate the study of impacts on the functioning of specific interorganizational networks types. In this case, industry benchmarks will also be used, thanks to which it will be possible to confront the scope and scale of the macroeconomic factors’ impacts—in the circumstances of an economic virus—on specific types of networks; e.g., comparing the effectiveness of oncology networks in selected countries with networks involving entities from the IT sector or networks within the automotive industry. Consequently, thanks to multi-faceted comparisons, it should be possible to identify both the negative and positive effects of the economic virus on organizations operating in interorganizational networks.

## Figures and Tables

**Table 1 ijerph-20-01178-t001:** Number of survey respondents representing individual units of the National Oncology Network in Poland.

Entity	Number of Survey Respondents
Maria Skłodowska-Curie National Research Institute of Oncology/Warsaw	1
Holy Cross Cancer Center/Kielce	2
Greater Poland Cancer Centre (GPCC)/Poznan	1
Maria Skłodowska-Curie National Research Institute of Oncology, Branch in Gliwice	1
Maria Skłodowska-Curie National Research Institute of Oncology, Branch in Krakow	2
West Pomeranian Oncology Centre/Szczecin	2
Copernicus Provincial Multidisciplinary Centre of Oncology and Traumatology/Lodz	2
Independent Public Health Care Facility of the Ministry of Internal Affairs with Warmia and Mazury Oncology Centre/Olsztyn	1
Provincial Clinical Hospital No. 1 named after Fryderyk Chopin/Rzeszow	2
Oncological Centre of the Specialist Hospital in Brzozow Podkarpacki	3
Lower Silesian Oncology Center/Wroclaw	2
Oncology Centre in Bydgoszcz	3
University Clinical Centre/Gdansk	2
Military Institute of Medicine/Warsaw	2
Beskidzkie Oncology Center/Bielsko-Biala	1
Pomorskie Hospitals/Gdynia	3
University Hospital of Karol Marcinkowski in Zielona Gora	1
Mazovian Provincial Hospital named after Saint John Paul II in Siedlce	2
Maria Sklodowska-Curie Bialystok Oncology Centre/Bialystok	2

Source: authors’ elaboration.

**Table 2 ijerph-20-01178-t002:** Summary of the survey questionnaire questions assigned to each explanatory variable (domains of relationships between units in the NON).

Domain	Question Number (Q)	The Content of the Question	Number of Specific Sub-Questions
Exchange	Q6_2	Sharing and transferring resources	1
	Q7_2	Importance of sharing and transferring resources	8
	Q8_2	Frequency of sharing and transferring resources in the given categories	8
Engagement	Q6_1	Cooperation in the context of organizational and management functions	1
	Q7_1	Importance of relationships in the given categories	19
	Q8_1	Frequency of relations in the given functions	19
	Q11_1	Features of the relationship regarding the level of engagement	15
Reciprocity	Q11_2	Features of the relationship between the centers make it possible to establish the level of reciprocity	15
	Q21_1	Number of specialized units cooperating on the part of a given center	6
	Q21_2	Number of specialized units cooperating on the part of the partner center	6
	Q22	Number of administrative units cooperating on the part of a given center	18
	Q24	Number of administrative units cooperating on the part of the partner center	18

Source: authors’ elaboration, the original numbering of the questions was used.

**Table 3 ijerph-20-01178-t003:** Spearman rank correlation coefficients for questions 6_2a, 7_2a, and 8_2a.

Question	Q6_2a	Q7_2a	Q8_2a
Q6_2a	1.00	0.34	**0.69**
Q7_2a	0.34	1.00	**0.59**
Q8_2a	**0.69**	**0.59**	1.00

Source: authors’ elaboration, the original numbering of the questions was used.

**Table 4 ijerph-20-01178-t004:** Multiple linear regression coefficients for Q28a vs. Q6_2a, Q7_2a, and Q8_2a.

Explanatory Variable	b (Coefficients in a Linear Regression Equation)	*p* (Statistical Significance)
Q6_2a	0.88	0.507
Q7_2a	1.39	0.328
Q8_2a	0.32	0.884

Source: authors’ elaboration, the original numbering of the questions was used.

**Table 5 ijerph-20-01178-t005:** Spearman rank correlation coefficients for Q6_1a, Q7_1a, Q8_1a, and Q11_1.

Question	Q6_1a	Q7_1a	Q8_1a	Q11_1
Q6_1a	1.00	−0.04	**0.46**	0.20
Q7_1a	−0.04	1.00	**0.43**	0.15
Q8_1a	**0.46**	**0.43**	1.00	0.19
Q11_1	0.20	0.15	0.19	1.00

Source: authors’ elaboration, the original numbering of questions was used.

**Table 6 ijerph-20-01178-t006:** Multiple linear regression coefficients for Q28a vs. Q6_1a, Q7_1a, Q8_1a, and Q11_1.

Explanatory Variable	b (Coefficients in a Linear Regression Equation)	*p* (Statistical Significance)
Q6_1a	0.47	0.006
Q7_1a	0.19	0.373
Q8_1a	−0.04	0.906
Q11_1	0.25	0.247

Source: authors’ elaboration, the original numbering of questions was used.

**Table 7 ijerph-20-01178-t007:** Multiple linear regression coefficients for Q28a vs. Q11_2, Q21_1a, Q21_2a, Q22a, and Q24a.

Explanatory Variable	b* (Normalized Coefficients)	b (Coefficients in a Linear Regression Equation)	*p* (Statistical Significance)
Q11_2	0.24	0.47	0.145
Q21_1a	0.29	0.28	0.337
Q21_2a	0.22	0.21	0.471
Q22a	0.16	0.05	0.514
Q24a	−0.14	−0.06	0.587

Source: authors’ elaboration, the original numbering of questions was used.

## Data Availability

Not applicable.

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
