# Peer review of "Effectiveness of Network Relations in Poland during the Economic Crisis Caused by COVID-19: Interorganizational Network Viewpoints"

_ijerph, 2023, doi:10.3390/ijerph20021178_

Round 1

Reviewer 1 Report

The subject of the article is interesting, and it is linked to the objectives of the journal, however, there are some issues that have to be reconsidered.

For better visibility on databases, the authors are asked not to repeat among keywords the words/concepts included in the title of the article.

The Abstract is too long. Also, it should include the main findings and the possible implications.

The Introduction part is not well written, it is more a description of the fact that scrutinize on the main findings presented in the literature. There should be a section on literature review (empirical literature and theoretical/conceptual framework). From which studies in the literature was the model used in the study created? 

The logic flow is another big problem in this paper. You can see logic jump and flaws scatter all over the manuscript. These logic flow issues are severe in this paper. The authors are strongly suggested to plan each discussion cohesively and use transition words and phrases to connect each part.

Overall, the paper provides some valuable results. However, it should be carefully rewritten to deliver this value to readers

Author Response

Dear Reviewer,

Thank you for your very insightful and engaging comments, as well as the tips and recommendations contained in the review of our article. We have tried to analyze all the information carefully and have addressed each of them separately. We hope our answers are comprehensive and satisfactory.

Answers for the review:

The subject of the article is interesting, and it is linked to the objectives of the journal, however, there are some issues that have to be reconsidered.

  • For better visibility on databases, the authors are asked not to repeat among keywords the words/concepts included in the title of the article.

Answer: We have made the indicated changes, thank you for that tip.

  • The Abstract is too long. Also, it should include the main findings and the possible implications.

Answer: We have shortened the abstract as recommended.

  • The Introduction part is not well written, it is more a description of the fact that scrutinize on the main findings presented in the literature. There should be a section on literature review (empirical literature and theoretical/conceptual framework). From which studies in the literature was the model used in the study created? 

Answer: The research model is presented in part titled: Research assumptions of relationship effectiveness in an inter-organizational oncology network – the essence of network relation’s effectiveness.  We don’t think we can repeat that information in the introduction. The introduction deliberately structures the entire article - to introduce readers to the empirical argument carried out.

  • The logic flow is another big problem in this paper. You can see logic jump and flaws scatter all over the manuscript. These logic flow issues are severe in this paper. The authors are strongly suggested to plan each discussion cohesively and use transition words and phrases to connect each part.

Answer: We have made the indicated changes, thank you for that tip. Changes are shown at the paper.

  • Overall, the paper provides some valuable results. However, it should be carefully rewritten to deliver this value to readers.

Answer: We have made the indicated changes, thank you for that tip. Changes are shown at the paper.

Aleksandra Sus, Michał Organa, Joanna Hołub-Iwan

Reviewer 2 Report

This is a conceptual manuscript aimed at the “COVID and the Economics of Public Health” special issue of IJERPH. It investigates the effectiveness of health networks from multiple domains under the restrictions of Covid-19 pandemics. Research is staged in Poland with network coverage of 19 health institutions. Authors provide a good literature review along with strong introduction of fundamental issues. Writing style of the paper is very good.

Following points must be considered by the authors before publication:

- Title does not sound grammatically correct. Either “effectiveness of network relations…” (authors use this form within the text) or “network relations’ effectiveness….” should be used. “Interorganizational networks viewpoints” should be connected via “:” to main title.

- I suggest the revision of abstract. At its current state, first paragraph is more like introduction. A one-two sentence motivation along with 2nd paragraph seems enough. Authors should also refrain from using citations in the abstract.

-Description of the data (survey) is fuzzy (page 10, lines 420-440). I’m having hard time to understand the difference between survey versions. Summary of the questions is fine but I also suggest that the complete surveys to be uploaded to a cloud drive (if possible with data) and cited in the text.

- I suggest authors to refrain using “clear correlation” by just looking to R values. Also, p-values for the model would be more important than individual terms’ significances (ANOVA tables can be given as a whole).

- I don’t want to make harsh criticism on the sample size and randomization of the respondents but in order to be statistically meaningful, authors must present supportive arguments (how they collected, the way the respondents chosen, why limited sample size, some descriptive values  etc…)

- I appreciate the new concept of “economic virus” proposed by the authors. But in order to create a research impact, authors must present ways to extend these ideas to other countries or network types along with possible shortcomings and risks.

Minor point: Formatting of the tables is way too bad. This is maybe from word template but should be corrected.

Author Response

Dear Reviewer,

Thank you for your very insightful and engaging comments, as well as the tips and recommendations contained in the review of our article. We have tried to analyze all the information carefully and have addressed each of them separately. We hope our answers are comprehensive and satisfactory.

Answers for the review:

This is a conceptual manuscript aimed at the “COVID and the Economics of Public Health” special issue of IJERPH. It investigates the effectiveness of health networks from multiple domains under the restrictions of Covid-19 pandemics. Research is staged in Poland with network coverage of 19 health institutions. Authors provide a good literature review along with strong introduction of fundamental issues. Writing style of the paper is very good.

Following points must be considered by the authors before publication:

  • Title does not sound grammatically correct. Either “effectiveness of network relations…” (authors use this form within the text) or “network relations’ effectiveness….” should be used. “Interorganizational networks viewpoints” should be connected via “:” to main title.

Answer: We have corrected the title as recommended.

  • I suggest the revision of abstract. At its current state, first paragraph is more like introduction. A one-two sentence motivation along with 2nd paragraph seems enough. Authors should also refrain from using citations in the abstract.

Answer: We have shortened the abstract as recommended.

  • Description of the data (survey) is fuzzy (page 10, lines 420-440). I’m having hard time to understand the difference between survey versions. Summary of the questions is fine but I also suggest that the complete surveys to be uploaded to a cloud drive (if possible with data) and cited in the text.

Answer: The mentioned fragment was slightly corrected. The numbers of particular questions were unified and simplified. The text is now more intuitive. According to the suggestion of uploading the data set to a cloud drive – we are still working on the dataset under consideration, so we would prefer not to share it at this time.

  • I suggest authors to refrain using “clear correlation” by just looking to R values. Also, p-values for the model would be more important than individual terms’ significances (ANOVA tables can be given as a whole).

Answer: The term “clear correlation” was replaced by, hopefully, more appropriate terms. The overall conclusions regarding particular elements of the described model were slightly softened. 

  • I don’t want to make harsh criticism on the sample size and randomization of the respondents but in order to be statistically meaningful, authors must present supportive arguments (how they collected, the way the respondents chosen, why limited sample size, some descriptive values etc…)

Answer: We have clearly explained the sampling process. The exact explanation can be currently seen in the section 5 (Empirical research of National Oncology Network in Poland), on page 10, lines 419-429.

  • I appreciate the new concept of “economic virus” proposed by the authors. But in order to create a research impact, authors must present ways to extend these ideas to other countries or network types along with possible shortcomings and risks.

Answer: Thank you for that recommendation. We would like to do it in our next paper – we also thought about that, but the subject is not easy, especially in the analysed (oncological) context in Polish conditions. We were thinking about countries which are near Poland – about Germany, Czech Republic and France. An explanatory text fragment was added to the section 6 (Conclusions), on page 16, lines 667-685.

  • Minor point: Formatting of the tables is way too bad. This is maybe from word template but should be corrected.

Answer: We have tried to correct that issue, and we hope now is much better. 

Aleksandra Sus, Michał Organa, Joanna Hołub-Iwan

Round 2

Reviewer 1 Report

The authors did answered to the concerns I had. The manuscript could be published.